# Protocol Report on the Transcranial Photobiomodulation for Alzheimer’s Disease (TRAP-AD) Study

**DOI:** 10.3390/healthcare11142017

**Published:** 2023-07-13

**Authors:** Dan V. Iosifescu, Xiaotong Song, Maia B. Gersten, Arwa Adib, Yoonju Cho, Katherine M. Collins, Kathy F. Yates, Aura M. Hurtado-Puerto, Kayla M. McEachern, Ricardo S. Osorio, Paolo Cassano

**Affiliations:** 1Nathan S. Kline Institute for Psychiatric Research, Orangeburg, NY 10962, USA; kate.collins@nki.rfmh.org (K.M.C.); kathy.yates@nyulangone.org (K.F.Y.); ricardo.osorio@nyulangone.org (R.S.O.); 2Department of Psychiatry, New York University Grossman School of Medicine, New York, NY 10016, USA; xiaotong.song@nyulangone.org (X.S.); arwa.adib@nyulangone.org (A.A.); 3Department of Psychiatry, Division of Neuropsychiatry, Massachusetts General Hospital, Boston, MA 02129, USA; mgersten@mgh.harvard.edu (M.B.G.); ycho9@mgh.harvard.edu (Y.C.); ahurtado@mgh.harvard.edu (A.M.H.-P.); kmceachern2@mgh.harvard.edu (K.M.M.); pcassano@mgh.harvard.edu (P.C.); 4Department of Psychiatry, Harvard Medical School, Boston, MA 02115, USA

**Keywords:** Alzheimer’s disease (AD), transcranial photobiomodulation (t-PBM), neuromodulation, cognition, functional MRI (fMRI), Phosphorus magnetic resonance spectroscopic imaging (31P-MRSI), Positron emission tomography (PET), laser

## Abstract

Background: Alzheimer’s disease’s (AD) prevalence is projected to increase as the population ages and current treatments are minimally effective. Transcranial photobiomodulation (t-PBM) with near-infrared (NIR) light penetrates into the cerebral cortex, stimulates the mitochondrial respiratory chain, and increases cerebral blood flow. Preliminary data suggests t-PBM may be efficacious in improving cognition in people with early AD and amnestic mild cognitive impairment (aMCI). Methods: In this randomized, double-blind, placebo-controlled study with aMCI and early AD participants, we will test the efficacy, safety, and impact on cognition of 24 sessions of t-PBM delivered over 8 weeks. Brain mechanisms of t-PBM in this population will be explored by testing whether the baseline tau burden (measured with ^18^F-MK6240), or changes in mitochondrial function over 8 weeks (assessed with ^31^P-MRSI), moderates the changes observed in cognitive functions after t-PBM therapy. We will also use changes in the fMRI Blood-Oxygenation-Level-Dependent (BOLD) signal after a single treatment to demonstrate t-PBM-dependent increases in prefrontal cortex blood flow. Conclusion: This study will test whether t-PBM, a low-cost, accessible, and user-friendly intervention, has the potential to improve cognition and function in an aMCI and early AD population.

## 1. Introduction

Alzheimer’s Disease (AD) affects an estimated 6.2 million Americans aged 65 and older [1]. As life expectancy continues to increase, the number of expected AD patients is predicted to reach 13.8 million by 2060 [1]. Furthermore, medical costs associated with AD are expected to surpass $1 trillion by 2050 [2]. Despite many years of clinical trials, few treatments have shown any clinical benefit beyond moderate symptom alleviation. Thus, it is imperative to investigate new therapeutic strategies that are clinically beneficial and more easily accessible for patients with AD.

Research over the last decade indicates that AD has a very long preclinical phase, lasting possibly >10 years [3] with mild cognitive impairment (MCI) being a potential transitional state from normal cognition to mild dementia due to AD [4]. Therefore, more emphasis has been placed on understanding the factors that influence early brain pathology in order to slow the ultimate clinical expression of AD. For many years, the dominant theory for the etiology of AD was based on the neurotoxic effects of amyloid aggregates (i.e., the ‘amyloid-cascade hypothesis’). This hypothesis posits that accumulation of amyloid beta (Aβ) peptides, a main component of plaques, in the brain parenchyma is the causative event in the pathogenesis of AD [5]. However, conflicting empirical observations have cast doubt over the validity of some aspects of this hypothesis. For example, amyloid plaques have been found to occur in cognitively normal individuals [6]. There is also only a weak correlation between the density of plaques and the degree of cognitive impairment [6,7]. Taken together, these findings provide little insight into the etiology of early amyloid deposition or the onset of clinical symptoms.

Several theories have been proposed to explain the limitations of the amyloid-cascade hypothesis. One is the mitochondrial cascade hypothesis in which inherited and environmental factors influence mitochondrial function. These factors are thought to lead to problems in adenosine triphosphate (ATP) production, and to the formation of amyloid beta and tau proteins (a microtubule-associated protein that is hyperphosphorylated and aggregated in AD) [8] that are believed to play a critical role in AD pathogenesis [9,10]. A more contemporary view suggests a neuroenergetic hypothesis, focusing on not only mitochondrial function, but overall energy production from glucose in the Central Nervous System (CNS). This theory postulates that a decrease in glucose crossing the blood–brain barrier leads to energy deficiency stress in the CNS, which in turn leads to the formation of amyloid beta and tau proteins, as well as AD progression [11]. In support of these theories, a decrease in glucose metabolism and mitochondrial activity has been noted decades before clinical symptoms of AD emerge [12]. Furthermore, in patients with AD, mitochondria are characterized by lowered oxidative phosphorylation, decreased ATP production, and increased reactive oxygen species [13].

Given these findings of increased oxidative stress in mitochondria of patients with AD, researchers have tried various ways to augment mitochondrial antioxidant defenses with dietary supplements. Multiple research groups administered antioxidants, such as selegiline, carotenes, and vitamins C and E, in patients with AD or healthy subjects and followed their cognitive function over time [14,15,16,17,18]. However, findings from these studies have been inconclusive. This may be because there are difficulties with finding adequate antioxidant dosages and combinations [19]; it is also possible that these compounds poorly penetrate the blood–brain barrier and may not adequately reach brain cell mitochondria [20]. Therefore, the discovery of more effective and targeted interventions for mitochondrial abnormalities in AD is of great importance.

Transcranial photobiomodulation (t-PBM) is a novel, non-invasive, neuromodulation intervention with the potential to become an at-home, wearable treatment for AD [21]. It uses non-retinal exposure to specific wavelengths, either visible red light (600 to 700 nm) or Near-Infrared Radiation (NIR, usually 810–1100 nm), to stimulate, heal, and repair damaged or dying cells and tissue. t-PBM with near-infrared (NIR) light, at wavelengths of 800~1200 nm, penetrates robustly into the cerebral cortex [22,23,24]. It has been found to act on the brain in two specific ways: (1) by increasing mitochondrial ATP production, and (2) by increasing regional Cerebral Blood Flow (rCBF). Specifically, NIR delivered through t-PBM is absorbed by a mitochondrial enzyme and chromophore, Cytochrome C Oxidase (CCO), and is only minimally dissipated as thermal energy [25,26]. Results in cellular and animal models indicate that t-PBM can enhance mitochondrial activity, because NIR delivers energy to the CCO and stimulates the mitochondrial respiratory chain, leading to increased ATP production [25,27,28]. This is relevant for AD, which is associated with hypometabolism in specific brain areas [29,30] and mitochondrial dysfunction [13,31]. In addition, NIR can improve mitochondrial activity by promoting the dissociation of nitric oxide (NO) from the CCO, releasing the binding site for oxygen and restoring oxidative phosphorylation [26]. The released NO may also act as a local vasodilator [32], resulting in a *focal*, increased rCBF [33]. A study on isolated mitochondria also reported increased RNA and protein synthesis after irradiation with a low-level laser (632.8 nm) [34].

Animal research suggests that t-PBM might exert, via its impact on mitochondria, beneficial effects on several pathophysiological mechanisms implicated in AD, such as oxidative stress [35,36], neuroinflammation [37,38], and deficits in neuroplasticity and brain-derived neurotrophic factor (BDNF) [39,40,41]. NIR can induce short bursts of reactive oxygen species (ROS), leading to the activation of antioxidant mechanisms and the activation of the transcription factor nuclear factor κB (NF-κB), resulting in decreased overexpression of the inducible form of nitric oxide synthase (iNOS) and in a reduction in oxidative stress [32,42,43]. In animal models, NIR light with 600 to 1000 nm reduced neuroinflammation by decreasing proinflammatory cytokines such as IL-6, TNF-α, IL-1β, and IL-8 [44,45,46], decreasing the infiltration of macrophages, and activating microglia and T lymphocytes to the CNS [46]. Studies have also shown that t-PBM stimulates neurogenesis and neuroprotective mechanisms in neuronal injury, possibly mediated by increased BDNF and by the inhibition of GSK-3β and pro-apoptotic molecules [32,47,48,49,50,51,52,53,54].

In addition, the use of t-PBM in animal models of AD has demonstrated efficacy in the reduction of Aβ burden in the brain. In one study using a mouse model of AD, NIR t-PBM was delivered to the tibia to stimulate bone marrow and mesenchymal stem cells. This was then associated with a 35% increase in the phagocytosis of Aβ and significant reduction in the Aβ brain burden [55]. In a study using TASTPM AD mice, 5-month treatment (with two treatment sessions per week) of 1072 nm NIR light delivered to the whole body resulted in an increase in heat-shock proteins (involved in maintaining healthy neurons) and a decrease in Aβ-associated proteins and plaques in the cerebral cortex, along with a reduction in tau-P [56]. De Taboada et al. demonstrated that t-PBM applied three times/week for 6 months at various irradiance doses significantly reduced Aβ plaques in Aβ protein precursor transgenic mice, and that there was a dose-dependent effect on the amyloid load [57]. t-PBM was also found to reduce the expression of inflammatory markers, increase ATP levels and mitochondrial function, and mitigate behavioral effects of advanced amyloid deposition [57]. In a similar case, Purushothuman et al. applied 670 nm NIR transcranially to two mouse models of dementia, the APPswe/PSEN1dE9 (APP/PS1) transgenic mouse engineered to develop amyloid plaques and the K369Itau (K3) transgenic mouse engineered to develop neurofibrillary tangles. Treatments applied 5 days a week for 4 weeks resulted in a decrease in the Aβ burden in the cerebellum of the APP/PS1 mice and in a reduction in immunoreactivity of hyperphosphorylated tau in the K3 mice compared to the sham [58]. Oxidative stress was also reduced in the K3 mice while the expression of CCO was maintained at near wild-type levels [58].

The safety and efficacy of t-PBM in humans have been demonstrated in studies across a variety of conditions, thereby making t-PBM a potential treatment strategy warranting further examination. In a pooled sample of 1410 subjects with stroke from three randomized control studies (RCTs) [59,60], no significant difference in the rate of adverse effects was observed between the groups receiving laser NIR (808 nm; 5 W) and sham. A clinical trial, and its replication with 16 sessions, reported an increased number of mild side-effects in the active treatment group, with the most frequent being insomnia, “seeing vivid colors”, “an ashtray-like taste”, and an irritable mood [33,61]. A pilot study of t-PBM in humans with cognitive deficits has shown promising results: 19 participants with impaired cognition were randomized to active or sham treatments over 12 weeks [62]. Active participants with moderate to severe impairment (Mini-Mental State exam [MMSE] scores 5–24) showed a significant improvement at the conclusion of treatment. In a case series of five patients with mild to moderate AD treated with intranasal home-use PBM devices, using pulsed, 810 nm light-emitting diodes (LEDs), a significant improvement was noted on neurocognitive tests such as MMSE and the Alzheimer’s Disease Assessment Scale–Cognitive Subscale (ADAS-Cog) [63].

## 2. Materials and Methods

### 2.1. Objectives

Taken together, t-PBM has great potential to serve as a safe and easily accessible treatment for aMCI and mild dementia due to AD. We hypothesize that compared to sham, t-PBM will be associated with an improvement in cognitive function (which also could manifest as a deceleration of rates of decline in cognitive function).

Aim #1: To assess the efficacy of t-PBM for cognitive deficits in amnestic MCI (aMCI) and mild dementia due to AD. Participants will be randomized to t-PBM or sham and their cognitive functions will be monitored with a change in the Repeatable Battery for the Assessment of Neuropsychological Status (RBANS-Update) [64] Total Scale score after 8 weeks of treatment;Aim #2: To evaluate the safety, tolerability, and feasibility of t-PBM treatment in patients with aMCI and mild dementia due to AD. We will monitor changes in treatment emergent adverse events by using Systematic Assessment for Treatment Emergent Events (SAFTEE) [65], as well as scales that assess suicidal ideation and sleep quality;Aim #3: To explore brain mechanisms of t-PBM in aMCI and mild dementia due to AD. We will assess possible mediators of the treatment response including baseline tau burden (measured using Positron Emitting Topography [PET] with a ^18^F MK-6240 tracer), changes in measures of mitochondrial function (assessed using Phosphorus Magnetic Resonance Spectroscopic Imaging [^31^P-MRSI]), and blood flow in the prefrontal cortex (PFC) (measured with a Blood-Oxygen-Level-Dependent [BOLD] signal change using functional Magnetic Resonance Imaging [fMRI]).

### 2.2. Study Design

The present study will be the first to evaluate the effects of t-PBM in aMCI and mild dementia due to AD in a parallel group, sham-controlled, 8-week randomized, multi-site clinical trial. The trial will run simultaneously across three study sites: NYU Langone Health (NYU), Nathan S. Kline Institute for Psychiatric Research (NKI), and Massachusetts General Hospital (MGH). With this data, we will attempt to assess the efficacy and safety of t-PBM as an easily administered treatment for those suffering from aMCI and mild dementia due to AD. This study has been registered on ClinicalTrials.gov (Identifier: NCT04784416). A schematic of the study design can be found in Figure 1.

### 2.3. Study Participants

In total, 125 participants, aged 65–85 (inclusive), will be recruited from three study sites, NYU, NKI, and MGH who meet the Petersen MCI criteria for aMCI [66], extended to also include early AD, with a Clinical Dementia Rating (CDR) [67] score between 0.5 and 1.0, and a Functional Assessment Staging (FAST) [68] score of 1–4. Participants will be randomized to either active t-PBM or sham treatments, lasting for 8 weeks.

### 2.4. Screening Procedures

After informed consent, participants will undergo full medical and psychiatric screening, as well as initial neuropsychological testing. This screening phase includes obtaining demographic information and administering the neuropsychological test screener Addenbrooke’s Cognitive Examination-III (ACE-III) [69], Clinical Global Impressions Scale-Severity (CGI-S) [70], Mini International Neuropsychiatric Interview (MINI) [71,72], MRI safety checklist, and Columbia Suicide Severity Risk Scale (C-SSRS) [73]. CDR, a measure that determines the severity of dementia on a 0–3-point scale, and FAST, a measure of dementia that assesses individual ability to function and perform activities of daily living, will also be completed. The New Immigrant Survey–Skin Color Scale (NIS-SCS) [74] will be administered, as t-PBM absorption may be dependent on skin tone. Medical history as well as concomitant medications and therapies will be reviewed by a clinician to determine eligibility. Peripheral blood and urine samples will be collected for laboratory testing and/or storage; clinical laboratory evaluations include hematology, biochemistry, and urinalysis. A portion of the collected blood will be stored as biospecimen and used for the extraction of DNA and the assessment of Apolipoprotein E (APOE) genotypes. Simoa assays will be used for the blood detection of tau, Aβ40/42, neurofilament light chain (NfL), and Glial Fibrillary Acidic Protein (GFAP). Participants are also asked to complete self-report measures, which include the Geriatric Depression Scale (GDS) [75], Anxiety Symptoms Questionnaire (ASQ) [76], irritability section of Concise Associated Symptom Tracking (CAST-IRR) [77], Pittsburgh Sleep Quality Index (PSQI) [78], and Quality of Life in Neurological Disorders (Neuro-QoL) [79].

### 2.5. Baseline Data

To elucidate t-PBM’s mechanisms of action, participants will undergo neuroimaging related to critical features of AD prior to study treatment: a tau load (^18^F MK-6240 PET), measures of brain bioenergetics (^31^P-MRSI), and functional connectivity (rs-fMRI). MRI scanning will begin with the acquisition of structural images. Participants will be asked to perform a 30 min dynamic ^18^F MK-6240 tau PET-MR scan 85 min after tracer injection, which involves a single IV bolus injection of 185 MBq of the ^18^F MK-6240 tracer. PET images corresponding to 90–110 min of post-injection tracer uptake will be acquired using a Siemens 3T PET/MR scanner. ^31^P-MRS acquisition (which takes less than 30 min) will take place in a separate session after relocation to a different MRI scanner. Subsequently, we will collect approximately 30 min of functional MRI data during the administration of a single, unblinded session of NIR t-PBM: ~10 min of fMRI before t-PBM, ~10 min coinciding with t-PBM, and ~10 min following t-PBM.

We will conduct additional neuropsychological testing in order to gather comprehensive baseline cognitive data for each participant. These test batteries include the North American Adult Reading Test-short version (NAART 35) as a measurement of premorbid verbal IQ [80,81], Letter and Pattern Comparison Task (LPC) as a measurement of attention and concentration [82], Face-Name Associative Memory Exam Short Form (FNAME-12) as a measurement of face recognition and memory [83,84], Trial Making Task (TMT) A and B as well as Stroop Color Word Test as measurements of processing speed and executive functioning [85,86], RBANS-Update as this study’s primary outcome measure, Letter Number Sequencing (LNS) as a measurement of working memory [87], and National Alzheimer’s Coordinating Center (NACC)’s Uniform Data Set Neuropsychological Battery Version 3 (UDSNB 3.0) [88]. If a participant is eligible for and interested in this study, their ACE-III scores during screening will also be used as a part of the comprehensive baseline cognitive data.

### 2.6. Randomization

After the collection of baseline data, participants will be randomized to either the active t-PBM group or the sham/control group. To protect study blinding, three lists of randomization sequences will be generated (one list for each site) using a block randomization of size 6, with three assignments to each of the two treatment groups within each block. The lists will be created via the “blockrand” package in R by a study statistician, who will not be involved in conducting any study procedures. Neither the study team nor the participants will know which number corresponds to the active t-PBM group, and the t-PBM device will not reveal any parameter settings of the treatment groups (i.e., only group numbers are displayed).

### 2.7. t-PBM Treatment Phase

Once randomization is complete, participants will complete t-PBM/sham treatments for ~12 min per day, 3 days per week, for 8 weeks. t-PBM will be delivered using laser probes placed over the forehead bilaterally (at standard electroencephalography (EEG) electrode positions F4, F3), via pulsed waves at an 808 nm wavelength. The specific parameters of administered t-PBM will be as follows: wavelength: 808 nm; average irradiance: 277.8 mW/cm^2^; peak irradiance: 833 mW/cm^2^; PW: 42 Hz; duty cycle: 33%; total area illuminated: 24 cm^2^ (2 × 12 cm^2^); exposure time: 667 s (11:07 min); fluence: 185.3 J/cm^2^; and total energy per session: 4.4 kJ. All treatments will be conducted in a clinic, and financial compensation for time and transportation will be provided to participants to minimize dropout. With the short duration of treatments and study staff’s expertise in retaining patients from the target population in our studies, we expect that the dropout rate will not exceed 20%.

Transcranial PhotoBioModulation-1000 (t-PBM-2.0) used in this study is an investigational device based on LiteCure’s LightForce^®^ EXPi Deep Tissue Laser Therapy^TM^ System. For our study, the EXPi System’s beam delivery (i.e., Empower^TM^) will be modified to non-invasively deliver NIR at 808 nm to the forehead. The modified system will also be configured to provide sham (placebo) treatment (with device modalities identical to t-PBM), which delivers NIR at less than 0.5% of the intensity of the active dose (in order to make the device functionality indistinguishable from active). No t-PBM light is reaching the eyes (which are also protected with polarized laser safety eyewear).

The t-PBM-2.0 device is considered a Class II medical device per 21 CFR 890.5500 and 878.4810, and is manufactured per 21 CFR 820. Each t-PBM-2.0 device consists of a therapeutic laser console (which produces laser energy as NIR), and an optical delivery system consisting of a flexible, double-sheathed optical fiber connected to a custom helmet (cap). The helmet is configured to deliver NIR light to EEG sites F4 and F3 (or in close proximity if covered by hair), covering a total surface treatment area of approximately 24 cm^2^ (12 cm^2^ × 2). The device utilizes a laser diode source with a maximum Continuous Wave (CW) output of ≤30 watts at a wavelength of 808 nm and nominal beam diameter of 40 mm at the outside aperture. It also includes laser safety eyewear with an optical density rating of >5.0 at 808 nm. Figure 2 showcases the t-PBM-2.0 treatment device.

The t-PBM-2.0 device operates in one of five modes: NIR continuous low irradiance (50 mW/cm^2^), NIR continuous middle irradiance (300 mW/cm^2^), NIR pulsed middle irradiance (300 mW/cm^2^), NIR continuous high irradiance (700 mW/cm^2^), and sham. Since the proposed average irradiance dose of t-PBM at ~300 mW/cm^2^ is the most frequently used dose reported in the literature [89], and has been associated with changes in CBF [90] and cognition [62,63], for this study we will only employ the NIR pulsed middle irradiance (~300 mW/cm^2^) and sham modes, at 42 Hz, with a pulsing 33% duty cycle. The 808 nm NIR wavelength was chosen for this study because it is at the absorption peak of the primary photoacceptor CCO [47,91], has optimal penetration through the skull [22,23,24], and has proven effects on CBF [90] and on cognition [62,63]. A laser source of NIR will be used in this study because lasers are currently the gold standard for t-PBM and are likely to deliver a superior therapeutic benefit compared to LEDs. Granted, LEDs can cover a larger brain area simultaneously and are cost-effective with a great safety profile, thus potentially easier to be self-administered at home [92]. However, one of the main issues with t-PBM is that a large proportion of the light administered to the scalp does not reach the brain [24]. Optimizing light penetration through the skull is very important, and LEDs have lower penetrance compared to lasers [22]. This is probably due to the higher coherence of light in lasers, compared to LEDs [93]. The anatomical sites targeted by t-PBM in this study include the bilateral dorsolateral prefrontal cortex (dlPFC) (standard EEG electrode sites F4, F3). We chose to broadly irradiate and engage the PFC because these areas are involved in relevant cognitive processes [94], and abnormalities in CBF have been noted in these areas of patients with AD [95,96]. Due to the nature of NIR t-PBM treatments and irradiance parameters, a warming sensation is identified as a potential concern for treatment tolerability. Although in many previous studies, including in AD, t-PBM with an irradiance of approximately 300 mW/cm^2^ was found to be tolerable, warming sensations will be monitored at each treatment session via the use of side-effect questionnaires.

Once per week during the treatment phase, participants will be asked to complete self-report measures (ASQ, GDS, Neuro-QoL, SAFTEE), review concomitant medications and therapies with study clinicians (who administer C-SSRS, Clinical Global Impressions Scale-Improvement (CGI-I), and CGI-S), have vital signs recorded, and report any adverse events. Once per month, participants will be asked to complete the PSQI and the Perceived Blinding Questionnaire (PBQ) [97].

### 2.8. Post-Study Data

Following the 8 weeks of t-PBM/sham treatment, participants will be asked to repeat the self-report measures mentioned previously. They will also be administered the C-SSRS, CGI-I, CGI-S, CDR, and FAST, and re-administered the neuropsychological test battery (different versions of the same batteries are used when possible to minimize practice effects). Their concomitant medications and therapies will be reviewed as well, with vital signs and adverse events recorded. Additionally, participants will undergo a blood draw and a ^31^P-MRSI scan as per the primary outcome procedures. Within 2 weeks of the primary outcome phase, participants will be asked to complete a short-term follow-up visit. This visit includes the completion of aforementioned self-report measures, as well as the administration of C-SSRS, CGI-I, and CGI-S; a review of concomitant medications and therapies, as well as recording of any adverse events, will also occur. Roughly 3 months after the completion of the primary outcome phase, participants will be asked to complete a long-term follow-up visit. This visit includes the completion of self-report measures, the administration of C-SSRS, CGI-I, and CGI-S, a review of concomitant medications and therapies, as well as a recording of any adverse events. Participants will also be re-administered the neuropsychological test battery, so data can be collected on whether or not the treatments’ effects on cognition (if any) persist for 3 months post treatment.

### 2.9. Data Management and Analysis

We hypothesize that compared to sham, t-PBM will be associated with an improvement in cognitive function (or with a deceleration of rates of decline in cognitive function), as measured with the primary outcome measurement (RBANS-Update Total Scale Score) assessed at the 8-week timepoint. There will be two datasets in this study, (1) an Intention-To-Treat (ITT) dataset and (2) a Modified Intention-To-Treat (MITT) dataset. ITT will include all randomized participants and MITT will include all participants who will undergo at least 1 week of t-PBM sessions and have at least one post randomization assessment. The efficacy analyses will include participants receiving the neuropsychological assessments at primary outcome (i.e., treatment completers), as neuropsychological data cannot be measured repeatedly without practice effects, despite the use of alternate test versions when possible. The MITT dataset will be used for almost all other analyses, including the examination of demographic or clinical differences between subject groups that do and do not undergo at least one scan.

The distribution of all variables will be investigated using descriptive statistics prior to a statistical analysis. Outliers will be identified and further examined to ensure data integrity. Transformations of the variables will be employed, and non-parametric statistical methods will be used when outcomes do not conform to required distributional assumptions. The treatment groups will be described with respect to baseline demographic, clinical, and brain characteristics. All analyses will be adjusted for age, sex, and ApoE4 status. The three study sites will be treated as fixed effects in all analyses.

The primary efficacy outcome will be assessed using the difference from baseline to week 8 in the RBANS-Update Total Scale Score, which has well-established normative ranges and psychometric properties [64,98]. We hypothesize that compared to the sham, t-PBM will be associated with superior cognitive function scores at the 8-week timepoint (primary outcome). This hypothesis will be tested using linear regression, which models the post-treatment score on composite neuropsychiatric measures as a function of treatment assignment, while adjusting for the baseline score of the composite as well as for sex, age, ApoE4 status, and site. In addition to the formal hypothesis testing, we will provide 90% confidence intervals for the estimated t-PBM effect and will report its effect size; the effect size will be computed by dividing the effect estimated in the linear regression by the standard deviation of the outcome measure at baseline.

Additionally, we will estimate the effect of t-PBM on the secondary outcomes (e.g., individual tests within the RBANS-Update), to allow for references to published effects from other studies; the type I error rate in these secondary analyses will be controlled with the false discovery rate [99]. We will also explore the potential differential effects of the three study sites by including them in the model for the outcome interactions: site by treatment, site by baseline covariate, and site by treatment by baseline covariate. Here, we will estimate the site effects rather than perform a formal hypothesis test.

We will also seek to explore the potential mechanisms of action of t-PBM treatment by investigating the following questions: (a) whether the baseline tau burden measured with ^18^F MK-6240 PET, or changes in mitochondrial function measured with ^31^P-MRSI, moderates the effect of t-PBM (in comparison to the sham) on the efficacy outcomes, and (b) whether the acute effects of t-PBM on cerebral blood flow in the PFC measured via a BOLD signal change on fMRI mediate the effect of t-PBM on the efficacy outcomes. For these, exploratory analyses will employ standard moderation (including interactions between treatment and potential moderators in the model for the outcomes) and mediation (a path analysis, estimating direct and indirect treatment effects) approaches [100]. We will control for age, sex, ApoE4 status, site, as well as depressive symptoms and sleep quality measured with the PSQI, both at baseline and after treatment.

With regards to detectable effects, the effects that can be detected with 80% power of two-sided tests with significance level alpha = 0.05 (detectable effects), with a total of 125 subjects, will depend on the dropout rate and the proportion of variance in the outcome that is explained by covariates in the model. With the short duration of individual treatments and our expertise in retaining patients from the target population in our studies, we expect that the dropout rate will not exceed 20%. In our experience, the baseline levels often explain 40–50% of the variance in neuropsychological outcome measures after treatments. Under such conditions, this study will be able to detect an effect of a medium size (Cohen’s d = 0.64).

### 2.10. Demographic Data on Existing Participants

Participants are recruited from the greater New York City and Boston metropolitan areas, as well as from the area surrounding Orangeburg, New York. As of 8 March 2023, a total of 109 participants have been recruited in this study. Forty-three participants self-identify as female (39.4%), and fifty participants self-identify as male (45.9%). The participants have a mean age of 74.14 years, with the youngest participant being 65 and the oldest participant being 85. All participants completed at least 12 years of education. The majority of participants (n = 70, 64.2%) identify themselves as White, with fifteen participants (13.8%) identifying themselves as Black/African American, one participant (0.9%) self-identifying as Asian, and three participants (2.8%) self-identifying as having more than one race. A little less than half of the participants are married (n = 51, 46.8%), while other participants report being divorced (n = 18, 16.5%), never married (n = 9, 8.3%), or a member of an unmarried couple (n = 3, 2.8%).

## 3. Discussion

Given the ever-increasing prevalence of AD [1] and the lack of effective, easily accessible treatments, the need for novel treatment strategies is dire. t-PBM is an emerging neuromodulation therapy that has a potential to treat AD [21,25,27,28,33,34,42,43,44,45,46,47,48,49,50,51,52,53,54,55,56,57,58,62,63,89,90,91], with a favorable safety profile [33,59,60,61]. Our study will be the first to evaluate the effects of t-PBM in aMCI and mild dementia due to AD in a parallel group, sham-controlled, 8-week randomized, multi-site clinical trial. We aim to assess treatment efficacy via the administration of an array of comprehensive neuropsychological assessment batteries that measure multiple domains of cognition.

Since the commencement of study activities, we have made several changes to our initial study protocol. One major change concerns the inclusion criteria. Originally, our inclusion criteria focused primarily on aMCI (with a CDR of 0.5 and a FAST of 1–3). However, we decided to expand our enrollment to include the mild dementia range in addition to our original target of aMCI, thus widening the original CDR and FAST inclusion ranges (to 0.5–1 and 1–4, respectively). It is possible that in doing so, we may see a differential response to t-PBM based on the heterogeneity of conditions and disease severity. Another change is related to neuroimaging procedures. Given the age range of our target population (65–85), MRI contraindications (such as pacemakers, joint replacements, and claustrophobia) are common. As neuroimaging data will not be analyzed as primary outcome, but only for exploratory analyses, we decided to randomize participants who could not undergo neuroimaging procedures. We also anticipate some difficulties with our participants, due to this population’s potential issues with involuntary movement and maintaining a steady posture during scans, but we are hopeful that we will gather data sufficient for meaningful analyses regardless.

There are several potential ethical concerns with our study that are common to research projects performed with AD participants. First, we have a sham group, for which we do not expect to observe improvements in cognition, and participants in this group may in fact decline. In addition, while we are hopeful that the active t-PBM group will demonstrate cognitive improvements, the extent of improvements as well as the persistence of improvements long term are unknown. To these points, we implemented an open-label option that allows all interested participants, regardless of their randomized group assignment, to receive active, unblinded t-PBM treatments—given that they demonstrate cognitive impairments as determined with the achieved ACE-III total score that is obtained during post-treatment neuropsychological testing.

We are planning to randomize 125 patients over the course of 5 years. As it is a double-blinded study, we will not report on any outcome results until all study procedures are complete and the dataset is locked. However, it is possible that we may release some initial baseline clinical, imaging, or biomarker data, which is dependent on the number of participants that complete these procedures.

## 4. Limitations and Future Directions

It is worth mentioning that we have opted not to collect data on Aβ burden in the brain, either by way of Amyloid PET or Aβ40/42 biomarkers in the cerebral spinal fluid. The reason behind this decision is four-fold: (1) Amyloid PET may be less specific to AD than tau PET, as patients with AD seem to show an increased tau tracer uptake and a characteristic brain distribution on tau PET, which is not present in other dementias such as frontotemporal dementia or diffuse Lewy body disease (patients of which often show increased tracer uptake on Amyloid PET) [101]; (2) with rare exceptions, amyloidosis appears to be required for the detection of high levels of 3R/4R tau deposition with novel tracers such as ^18^F-MK6240 [102,103]; (3) it is possible for abnormal Amyloid PET results to be compatible with normal cognition, but the same is less likely for highly abnormal tau PET [6,102]; and (4) AD is normally characterized first by the appearance of amyloidosis and later by tauopathy, with tauopathy being the proteinopathy mainly associated with clinical symptoms, making tau PET more relevant for clinical symptom monitoring [102,103]. With that said, despite recent contentions in the field regarding the validity of the amyloid cascade hypothesis, measurements of Aβ burden in the brain remain widely used in clinical trials involving patients with AD. Given the logistical and safety difficulties of acquiring both Amyloid and tau PET and performing lumbar punctures, we decided to only obtain the tau PET (also for the reasons discussed above), complemented by the collection of peripheral blood samples for analyzing plasma Aβ40/42 levels.

The risks associated with this study include risks of the t-PBM and of the neuroimaging procedures utilized. t-PBM has a favorable safety profile as demonstrated by previous studies [33,59,60,61]. Neuroimaging procedures (especially PET, which includes a radioactive tracer) also carry a small safety risk. Extensive safety data will be collected (as described in Section 2) to monitor and evaluate these risks.

Future study directions may include the implementation of additional blood work. In upcoming studies, we plan to examine how central and peripheral mitochondrial metabolism contributes to cognitive performances in clinical phenotypes of prodromal AD and mild dementia due to AD in the context of t-PBM treatments. We will use novel exosome analysis techniques to determine in vivo molecular biosignatures of brain mitochondrial metabolism, with a focus on the mitochondrial metabolite acetyl-L-carnitine (LAC). Specifically, we will measure key markers of LAC-related mitochondrial metabolism in exosomes that are enriched for the L1 Cell Adhesion Molecule (L1CAM), a protein highly expressed in the brain, at the following three time points: prior to the beginning of t-PBM/sham, 1 week since the initiation of t-PBM/sham, and at the conclusion of t-PBM/sham (i.e., 8 weeks since treatment initiation). In addition, we will enhance our protocol to isolate exosomes enriched for specific brain areas/neurons of interest and characterize their molecular cargo. We will achieve this by characterizing the transcriptomic profiles in the response to t-PBM using unbiased RNAseq assays and bioinformatic approaches. Next, we will measure peripheral LAC levels and the related mitochondrial metabolism using mass-spectrometry. We will also assess peripheral insulin resistance, which is a metabolic dysfunction that is regulated via LAC-related pathways. Lastly, we will use computational approaches to determine the role of the novel biomarkers of t-PBM responses in trajectories of functional connectivity.

Future studies may also want to investigate the effect of t-PBM on brain tau burden. In the current study, we are only exploring whether the baseline tau burden predicts the t-PBM response and are not following up with any post-treatment PET scans. If our hypothesis is supported and t-PBM is found to improve memory and cognition in subjects with aMCI and mild dementia due to AD, future research may elucidate the mechanism of action of t-PBM by investigating changes in the tau or amyloid load as assessed with PET throughout a course of treatment.

## 5. Conclusions

We are describing a large study testing t-PBM, a novel neuromodulation strategy, as a potential treatment for aMCI and early AD. The importance of this study is three-fold: (1) it targets aMCI and early AD, which are important initial stages of AD that lack adequate approved treatments, (2) it evaluates the efficacy and safety of t-PBM, an innovative, non-invasive technology that has a well-established safety profile, for improving brain function and cognition at the aforementioned initial AD stages, and (3) it explores the association of t-PBM treatment effects with important biomarkers relevant for AD illness progression. If effects are confirmed, the present study will both support short-term clinical development of an easy-to-scale device for the treatment of aMCI and AD, and validate biomarkers for the development of future, novel modulation strategies.

## Figures and Tables

**Figure 1 healthcare-11-02017-f001:**
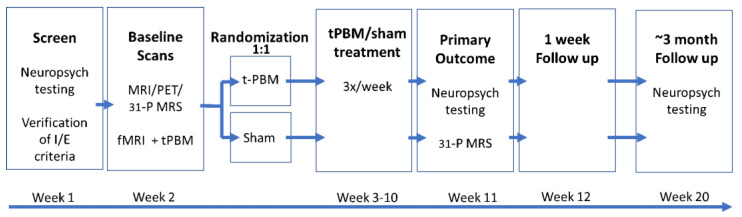
Schematic of TRAP-AD study design.

**Figure 2 healthcare-11-02017-f002:**
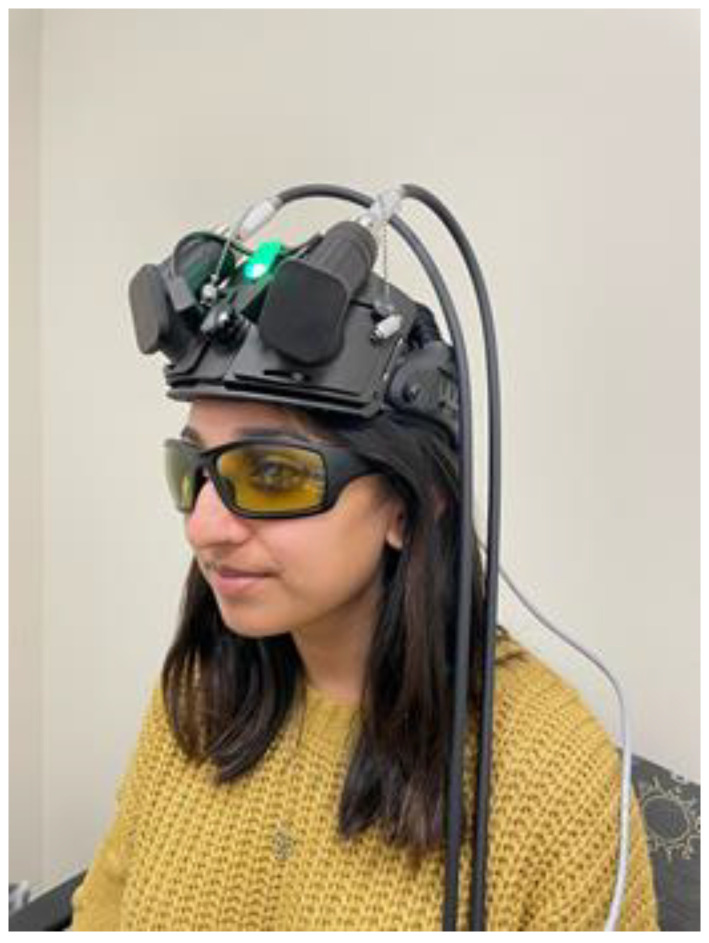
TRAP-AD study treatment device, t-PBM-2.0.

## Data Availability

Not applicable for this manuscript. The results of the completed study will be shared, in agreement with the data sharing policies of the National Institute for Aging.

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
