# Peer review of "Protocol Report on the Transcranial Photobiomodulation for Alzheimer’s Disease (TRAP-AD) Study"

_healthcare, 2023, doi:10.3390/healthcare11142017_

Round 1

Reviewer 1 Report

eview of “Protocol report on the Transcranial Photobiomodulation for Alzheimer's disease (TRAP-AD) Study”

Dear Authors,

Thank you for the opportunity to review this article. In this article, you have provided description of the planned research which is about to be made, t-PBM for Alzheimer's disease. A total of 24 sessions should be made over 8 weeks, on 125 participants. The planned research seems interesting and I would definitely like to read about the results.

This is the first time I am reviewing this type of article, describing protocol of some future research. I could not find any article of this type in this journal, for comparison. I have noticed that you have not followed recommended structure of this type of the text. You should restructure the text, to include Expected results and remove Discussion and Conclusion.

Also, literature is not correctly cited in the text nor in the list of references.

I recommend that the structure of the text should be revised according to instructions for this type of the paper.

Author Response

We have restructured the text and references using templates from protocol manuscripts previously published by Healthcare Journal. Keeping in line with the formats of published protocol manuscripts, we have kept the discussion and conclusion sections which state our hypothesis, discuss limitations, and explore future study directions. We have double-checked all references to ensure that they are relevant and formatted correctly.

Reviewer 2 Report

This is an ambitious and important effort to further validate the promising t-PBM claims in an area where patients cannot be offered too much hope. Below are a few comment on the text, for consideration.

-          The study is using laser, and for good reasons, considering compatability with MIR. Nevertheless, the literature is diffuse, with not only different wavelengths, energies and doses but also with LEDs or lasers. A few lines for the non-specialist reader could be helpful here to briefly describe the differences with pros and cons.

-          A photo of the unit in situ would be helpful

-          As for traditional PBM studies, more details are suggested regarding J/cm plus total J per session

-          Different PRRs are suggested in the literature, i.e. 40 Hz for a specific reason. Would be interesting to learn about the choice of 30% PRR, if not only for heat dissipation.

-          How was the sham device equipped? LEDs? 808 nm is only partly invisible to the human eye.

-          Page 6/275, ”dose” is used instead of ”irradiance”

-          The manuscript uses ”laser therapy” as well as ”low-level laser therapy, but ”PBM” is now the accepted term. Hard to remember.

-          Page 3/115. What does ”chronic treatment” mean?

-          Ref 3, minor editing

Author Response

  1. We added the following paragraph to our methods section (page 7, lines 292-300): “A laser source of NIR is used in this study because lasers are currently the gold standard for t-PBM and are likely to deliver superior therapeutic benefit compared to LEDs. It is true that LEDs can cover a larger brain area at the same time, are less expensive and very safe, thus easier to be used in self administration at home [100]. However, one of the main problems of t-PBM is that a large proportion of the light administered at the scalp does not reach the brain [101]. Optimizing light penetration through the skull is very important, and LEDs have lower penetrance compared to lasers [102]. This is probably due to the higher coherence of light in lasers, compared to LEDs [103].”
  2. We have provided a photo of the treatment device (Figure 2), on page 6, line 281.
  3. The specific parameters of t-PBM used were added page 6, lines 253-256: “The specific parameters of t-PBM administered are: wavelength: 808 nm, average irradiance: 277.8 mW/cm2, peak irradiance: 833 mW/cm2, PW: 42 Hz, 33% duty cycle; total area illuminated: 24 cm2 (2 x 12 cm2); exposure time: 667 s (11:07 min); fluence: 185.3 J/cm2; total energy per session: 4.4 kJ”
  4. As the reviewer points out, the optimal frequency of PW in t-PBM has not been determined. Our devices have a frequency of 42 Hz, which is very close to the 40 Hz found to be helpful in AD with other stimulation modalities. The frequency was listed among the specific parameters of t-PBM used (page 6)
  5. Addressed on page 6, lines 264-268
  6. We believe the word “dose” is helpful for readers not familiar with t-PBM to understand the variation in irradiance. We changed to “irradiance dose” (page 3, lines 118-119, citation #57) to address the reviewer’s concern.
  7. We agree and we changed the terminology to “t-PBM” consistently.
  8. Addressed on page 3, we changed “chronic treatment” to “5-month treatment (with two treatment sessions per week)”
  9. We edited the reference

Reviewer 3 Report

The paper presents a study protocol for t-PBM Photobiomodulation for Alzheimer’s Disease, it is potential treatment for AD.  Hope the team can be successful. Several points need to be clear:

1.  ....to deliver NIR light to EEG sites F4 and F3. It is better to mention why is the area;  and why choose 808nm light?

2. the light intensity used is 300mW/cm2 or higher, will it too much heat for subject? please mention the issue;

Author Response

  1. The rationales for choosing dlPFC (F4, F3) as our treatment target and for choosing 808 nm light are given on pages 6-7: “The 808 nm NIR wavelength was chosen for this study because it is at the absorption peak of the primary photoacceptor CCO [47,91], has optimal penetration through the skull [22-24], and has proven effects on CBF [90] and on cognition [62,63].” “The anatomical sites targeted by t-PBM include bilateral dorsolateral prefrontal cortex (dlPFC) (standard EEG electrode sites F4, F3). We chose to broadly irradiate and engage the PFC because these areas are involved in relevant cognitive processes [92], and ab-normalities in CBF have been noted in these areas of patients with AD [93,94].”
  2. Addressed on page 7, lines 304-308.

Reviewer 4 Report

This manuscript describes a planned clinical trial of transcranial PBM for MCI and early-stage dementia. As such, there are no data contained in the manuscript - just a description of the trial. The trial appears well designed, and will be very important for understanding the clinical utility of t-PBM in the context of dementia.

A few comments:

- I found it a bit strange that the authors plan to use PET but not do any Abeta imaging. While I agree that the amyloid cascade hypothesis is fundamentally flawed, a read-out of amyloid burden (and the effect of treatment on this) is a mainstay for almost all clinical trials of AD. Given the effort to recruit this many patients, it would seem a shame not to take the opportunity to collect these data.

- Has the trial been registered? If so, please include the registration number.

- How was the recruitment target of 125 determined? Please provide some detail of the power to detect a particular effect size with this sample.

- Will the t-PBM treatments be conducted in clinic or at home? If in clinic, are there any concerns about drop-out (given there will be 24 treatments over 8 weeks), and measures to mitigate this? If at home, are there any concerns about incorrect use, and measures to mitigate this?

- Lines 311 and 335 mention a hypothesis of "greater improvement". I wonder if there might be a more appropriate way to phrase this. I can't imagine there is any expectation that the sham group will improve? And it would still be a great result if there was a difference in change from baseline when comparing sham to t-PBM, even if there was no improvement (e.g. sham declined but t-PBM stabilised relative to baseline). Not a massive issue though... just something to consider.

Author Response

  1. Addressed on page 9-10, “Limitations” lines 447-464. “It is worth mentioning that we have opted not to collect data on Aβ burden in the brain, either by way of Amyloid PET or Aβ40/42 biomarkers in cerebral spinal fluid. The reason behind this decision is four-fold: 1) Amyloid PET is less specific to AD than tau PET. Patients with AD seem to show an increased tau tracer uptake and a characteristic brain distribution on tau PET, which is not present in other dementias such as frontotemporal dementia or diffuse Lewy body disease (patients of which often show increased tracer uptake on Amyloid PET) [103]; 2) With rare exceptions, amyloidosis appears to be re-quired for the detection of high levels of 3R/4R tau deposition with novel tracers such as 18F-MK6240 [100,101]; 3) It is possible for abnormal Amyloid PET results to be compatible with normal cognition, but the same is less likely for highly abnormal tau PET [6,104]; and 4) AD is normally characterized first by the appearance of amyloidosis and later by tauopathy, with tauopathy being the proteinopathy mainly associated with clinical symptoms, making tau PET more relevant for clinical symptoms monitoring [104,105]. With that said, despite recent contentions in the field regarding the validity of the amyloid cascade hypothesis, measurements of Aβ burden in the brain remain widely used in clinical trials involving patients with AD. Given the logistical and safety difficulties of acquiring both amyloid and tau PET, we decided to only get the later (for the reasons discussed) and to also collect plasma beta amyloid levels”
  2. Addressed on page 4, lines 177-178.
  3. We added a justification of the sample size in the Data analysis section (page 8, lines 385-392)
  4. Addressed on page 6, lines 257-260.
  5. This is a relatively short-term study, so it is unclear if cognitive deterioration should be expected in the absence of treatment (i.e., sham). However, the term “greater improvement” in cognition does also cover the possibility that sham will be associated with a decrease in cognitive function. However, we addressed the reviewer’s point on page 3, lines 150-151; page 7, line 337; and page 8, line 359.

Round 2

Reviewer 1 Report

Dear Authors, it is much easier to understand the manuscript with this changed structure and other changes which are made. I do not have further comments. I will recommend the manuscript to be accepted.